# Computer-Assisted Evaluation Confirms Spontaneous Healing of Donor Site One Year following Bone Block Harvesting from Mandibular Retromolar Region—A Cohort Study

**DOI:** 10.3390/diagnostics14050504

**Published:** 2024-02-27

**Authors:** Shadi Daoud, Adeeb Zoabi, Adi Kasem, Amir Totry, Daniel Oren, Idan Redenski, Samer Srouji, Fares Kablan

**Affiliations:** 1Department of Oral and Maxillofacial Surgery, Galilee College of Dental Sciences, Galilee Medical Center, Nahariya 2210001, Israel; dr.adeebz@gmail.com (A.Z.); adialkasem@hotmail.com (A.K.); amirtotry@gmail.com (A.T.); dannyoren100@walla.co.il (D.O.); idan.redenski@gmail.com (I.R.); dr.samersrouji@gmail.com (S.S.); kablanp1@gmail.com (F.K.); 2The Azrieli Faculty of Medicine, Bar-Ilan University, Safed 1311502, Israel

**Keywords:** bone healing, bone block, segmentation, volumetric analysis, Hounsfield units’ evaluation

## Abstract

Bone augmentation prior to dental implant placement is a common scenario in the dental implantology field. Among the important intraoral harvesting sites to obtain bone blocks is the ramus/retromolar region that has a high success rate and long-lasting alveolar ridge augmentation. Preserving the bone volume and quality at the donor site is crucial for preventing further complications or to serve as a site for re-harvesting. Healing of the intraoral donor sites has been described in the maxillofacial field. This study aimed to evaluate the spontaneous healing of the mandibular retromolar donor site utilizing computer-assisted quantification 6 and 12 months after bone harvesting. Materials and methods: The study was conducted on patients who underwent an alveolar ridge augmentation using an intraoral retromolar bone graft. Three CBCT scans were performed—intraoperative, and at six months and one year after the surgical procedure. By using the Materialise Mimics Innovation Suite software 26.0 features segmentation by thresholding, Hounsfield unit averaging, and superimposition of the tomographies, we could precisely quantify the healing process utilizing spatial and characteristic measures. Results: In all cases, the computer-aided quantification showed that six months following surgery, the donor site had recovered up to 64.5% ± 4.24 of its initial volume, and this recovery increased to 89.2% ± 2.6 after one year. Moreover, the Hounsfield unit averaging confirmed dynamic bone quality healing, starting at 690.3 ± 81 HU for the bone block, decreasing to 102 ± 27.8 HU at six months postoperatively, and improving to 453.9 ± 91.4 HU at the donor site after a year. Conclusions: This study demonstrates that there is no need for additional replanting at the donor site following retromolar bone block harvesting, whether autogenous or allograft, since spontaneous healing occurs 12 months following the surgery.

## 1. Introduction

The use of oral implants in dental rehabilitation has evolved into a common treatment modality, providing consistently reliable long-term results [1]. A sufficient bone height and width are crucial for a successful dental implant procedure [2,3]. In patients with compromised alveolar bone, bone augmentation is considered for the placement of dental implants with an adequate length and diameter for proper prosthetic function [4].

Several bone grafting materials and techniques have been used, including guided bone regeneration, sandwich technique, onlay blocks, and distraction osteogenesis.

Autogenous bone grafts are widely accepted as the gold standard due to their osteogenic, osteoinductive, and osteoconductive biological activities, in addition to their safety and their excellent incorporation at the recipient bed [5]. 

Autogenous bone grafts can be obtained from extraoral or intraoral donor sites. These harvest sites differ based on their embryological characteristics. Intraoral intramembranous bone grafts undergo minimal resorption [6,7]. The main intraoral harvest sites are the mandibular symphysis and mandibular ramus, with a high success rate for long-lasting alveolar ridge augmentation, up to complete jaw augmentation or extensive bone reconstruction [8,9,10]. 

The healing of an intraoral donor site after bone harvest and re-harvesting from the mandibular ramus and symphysis have been investigated and described in the maxillofacial literature. Schwarts et al. (2009) described the clinical and histologic features of new bone formation at symphysial donor sites and their re-harvesting potential [11]. In their clinical and tomographic study, Verdugo et al. (2010) described bone repair at the symphysis donor sites. They found bone repair rates of 79.8% 24 months following the harvesting [12]. Generally, osteogenesis at the donor site is facilitated by the integrity of the periosteum and the exposed medulla, serving as a vital source of osteoprogenitor cells.

In 2005, Pikos discussed the possibility of a second re-harvesting of a bone graft from the mandibular ramus following spontaneous healing [13]. Claudino et al. (2014) reported on a second bone graft harvest from the ramus area in one of their patients [14]. In addition, the authors have revisited the same retromolar area to obtain a bone block for further bone augmentation in some patients, according to unreported data.

Historically, measuring the volume and quality of healing donor sites relied on manual 2D assessments from CBCT cross-sectional sections [15]. However, recent advancements have introduced more efficient and precise methods. Advanced 3D imaging, improved precision, and automated tools now enhance the assessment of volume and quality, providing a significant upgrade from traditional approaches [16,17,18]. 

The aim of the current work was to quantitatively evaluate the spontaneous healing and recovery process at the mandibular retromolar donor site, employing computer-assisted techniques for detailed assessments.

## 2. Materials and Methods

The study involved 20 patients who underwent external oblique ridge bone block harvesting. The average age of the participants was 48.3 years, with a range of 37 to 62 years, and the group included 8 men and 12 women. All participants had unremarkable medical histories. Informed consent was obtained from each patient, following a thorough explanation of the procedures. The study received ethical approval from the Israeli Helsinki Committee system at the Galilee Medical Center in Naharya, Israel.

### 2.1. Pre-Operative and Follow-Up Radiological Evaluation

The patients’ treatment began with a preoperative CBCT assessment to evaluate the bone volume and quality at the donor and implantation sites to guide the treatment plan. A follow-up CBCT at six months post-surgery assessed the bone healing and graft site readiness for implantation [19]. A final CBCT was performed one-year post-surgery to assess the recovery at both the grafted and donor sites ([Fig diagnostics-14-00504-ch001]). In all our chosen cases, the grafted areas were in the posterior mandible, so the donor site was not the main reason for performing the CBCT at these time points, but by performing them we could assess the healing process at the donor site. CBCT images were acquired using a scanner (Planmeca ProMax^®^ 3D Classic, Helsinki, Finland) using one setting: 0.3 mm, 24 s, 106 kV, and 65 mAs.

### 2.2. Surgical Procedure

The intraoral bone harvesting for block grafting was typically carried out with local anesthesia and intravenous sedation. However, in cases of extensive reconstructions that involved multiple donor sites or surgeries, general anesthesia was recommended.

Preoperative antibiotics were given in accordance with established and accepted medical protocols [20]. In most cases, an inferior alveolar nerve block was avoided, and instead, only local vestibular and lingual infiltration with 2% lidocaine and 1:100,000 epinephrine (Septodont, PN #99167) was administered. This approach was chosen to ensure that minimal sensation remained, serving as a warning for the surgeon when they approached the inferior alveolar nerve. A trapeze-like incision, which commenced distal to the second molar with a 2 cm vestibular incision over the ramus bone, was made. Subsequently, a mucoperiosteal flap was raised, exposing the bone at the level of the external oblique ridge to a length of 3 to 4 cm and a depth of 2 cm [21]. The amount of bone to be harvested was determined based on the size and extent of the external oblique ridge and the amount of bone required for grafting. The harvesting osteotomy was conducted using the MicroSaw technique protocol [21]; a total of three osteotomies are performed using the diamond disk: two proximovertical osteotomies and one apicohorizontal osteotomy, as illustrated in Figure 1.

An additional osteotomy, positioned on the occlusal crest parallel to the external oblique ridge, was executed using a slender 1 mm drill bur. To facilitate the harvesting of the bone block, small perforations were created parallel to the buccal bone wall, reaching a depth of approximately 3 to 4 mm [21]. These perforations were located between the two vertical incisions mentioned previously. To prepare the bone block, these perforations were then connected using a fine chisel, inducing tension in the cortical bone. This technique generated a type of “explosive effect” around the crestal perforations, facilitating the easy lateral dislocation of the bone block, as shown in Figure 1. The donor site was closed using layered sutures, beginning with the reapproximating of the periosteal layer using 4–0 multifilament (Vicryl) sutures, and the grafting procedures were then carried out.

### 2.3. Intraoperative Bone Graft Scan

During the intraoperative phase, we employed a CBCT real-time scanning approach using a Planmeca ProMax^®^ 3D Classic CBCT scanner and its accompanying software using the following specifications: 8 × 8 cm^3^ field of view, 48 mAs, high contrast, 110 kV, slice thickness of 0.5 mm, image intensifier-based CT detector 12-bit grayscale. The total scanning time was 16 s. These settings allowed us to precisely assess the volume and quality of the bony graft (Figure 2).

### 2.4. Computer-Assisted Bone Healing Evaluation

In this study, computer-assisted evaluation was used to assess the healing process of the donor site based on the CBCT scans performed at different time points ([Fig diagnostics-14-00504-ch001]). Furthermore, the CBCT scans provided a means for precise bone density and quality measurements using the Hounsfield unit (HU) averaging feature. 

A computed evaluation of the process was performed using Mimics Innovation Suite software (Materialise, Leuven, Belgium) features. The process included three steps: segmentation by thresholding, superimposition of objects followed by Boolean subtraction for volume assessment, and Hounsfield unit averaging for bone density/quality evaluation.

### 2.5. Segmentation by Thresholding

Segmentation by thresholding involves isolating specific features or regions of interest in an image by setting a pixel intensity threshold, which allows for the precise delineation of structures based on their grayscale values [22]. In this study, Mimics software (Materialise, Leuven, Belgium) was utilized for segmentation in the analysis of four CBCT scans: preoperative, midterm, and one-year follow-up scans, in addition to the intraoperative graft scan. This process generated three-dimensional objects that were suitable for various applications within the research (Figure 2A and Figure 3A–C).

### 2.6. Superimposition of Objects and Boolean Subtraction

The superimposition of tomographies/3D objects, also known as image registration or image fusion, is a process in medical imaging where two or more tomographic images, such as computed tomography scans, are aligned and overlaid on top of each other to create a combined image [23]. This can be performed manually or through automated algorithms using specialized software.

The goal of superimposition of tomographies/3D objects is to accurately align and fuse multiple images from different time points to facilitate comparisons, analysis, and interpretation of the combined information. This application can be used for follow-up assessments: by comparing sequential CT scans taken at different time points, we can monitor changes over time, such as tumor growth, treatment response, or disease progression. Any differences or changes between the scans can be visualized and quantified by overlaying the images.

In this study, we utilized 3D objects obtained through the segmentation process for superimposition, relying on intact bony areas. Specifically, we conducted superimposition on segmented mandibles from preoperative and follow-ups scans by aligning intact bony regions, allowing for a comprehensive assessment of changes over time (Figure 3D). Employing 3D subtraction on the six-month mandibular object from the preoperative object revealed the residual bone defect (Figure 3E,F). Additionally, segmentation of the harvested bone block facilitated a comparison between residual bone volume and the grafted bone, enabling a detailed evaluation of the healing process (Figure 4A). The superimposition and volume assessments were conducted using 3-Matic software 18.0 (Materialise, Leuven, Belgium).

### 2.7. Hounsfield Unit Averaging

HU averaging is a technique employed in CT imaging to determine the mean HU value within a defined ROI or volume of interest. HUs represent a numerical assessment of the radiodensity observed in CT scans, offering insights into the composition, density, and potential pathology of anatomical structures and tissues. By assessing the HUs in particular regions, we can quantitatively assess the progress of bone healing in the donor site [18,24].

In this study, we employed a volumetric Hounsfield unit evaluation, beginning with the averaging of HUs in the harvested bone block scan, and then assessing the mean HU in the follow-up scans at the donor site. To ensure precision and consistency, we used the volume of the grafted bone block—following its superimposition back to its original location in the jaw (Figure 4A)—as a volumetric template for all subsequent volumetric HU evaluations (Figure 4B,C).

### 2.8. Statistical Analysis

In the current work, descriptive analyses of the data were represented as means ± standard deviation. A two-tailed paired sample t-test was performed to compare data across time points, with a significant level of 0.05. The analyses were performed using SPSS statistics software (version 27.0, IBM, Chicago, IL, USA).

## 3. Results

This study included 20 participants (8 males and 12 females; mean age: 48.3 ± 11.1 years) who underwent retromolar bone harvesting. The focus was on evaluating bone healing at the donor site through both volumetric and quality measures. There was no significant difference in the healing ratio based on pre-operative bone block size, gender, age, and involved jaw side. The intraoperative scanning results displayed the volume and density of the bone graft. The subsequent evaluations at 6 months and 12 months postoperatively provided insights into the healing process at the donor site. These assessments revealed a progressive healing process, culminating in near-complete healing of the donor site one year after the operation (Figure 5).

### 3.1. Bone Healing at Donor Site—Volumetric Evaluation

The total average volume of bone harvested was 606.5 ± 77.7 mm^3^, and ranging between 485.7 and 740 mm^3^. The residual bony defect showed a significant reduction over time (Table 1). On average, there was a 64.5% ± 4.24 healing in the bony defect at 6 months, and by one year postoperatively, this improvement was more pronounced, with an average healing of 89.2% ± 2.6 (Figure 6). These findings, graphically represented in histograms for each patient, underscored the effective healing at the donor site, illustrating a consistent trend of bone regeneration over the year following surgery (*p* < 0.05).

### 3.2. Bone Healing at Donor Site—Quality/Density Evaluation

The intraoperative HU average for the scanned bone grafts was 690.3 ± 81. At the 6-month follow-up, an expected lower average was observed (102.5 ± 27.8), demonstrating a decreased bone density at that timepoint in the donor site. However, by the 12-month follow-up, there was a substantial increase in the average HU to 453.9 ± 91.4 (*p* < 0.05) (data shown in Table 2 and Figure 7). These specific measurements indicate a notable increase in bone density and quality one year postoperatively.

## 4. Discussion

In oral implantology, the preference for intraoral sites, especially the mandibular ramus/retromolar area, for bone block grafts is widely recognized. This is due to their accessibility, lower associated morbidity, and effective outcomes [2,8,21].

Pikos in 2005 stated that harvesting of a bone block from the retromolar area can provide bone volume for three tooth segments [13]. The grafting of more sites may be performed with a longitudinal split of the harvested bone block (Khoury 2007) [25]. To increase the number of augmented sites utilizing the same bone block, Kablan described the advantages of a new method, the “Wedge Technique”, which involves multiple longitudinal and horizontal splittings of the harvested bone block to create multiple small and thin bone blocks (8–12 bone wedges) that can augment at least two recipient sites [26,27,28].

The surgical management of the donor site after bone block harvesting varies. Some teams opt to replant a portion of the bone block at the origin, reconstructing the external oblique line with micro-screws [21]. This, however, could reduce the available bone for augmentation at the recipient site, not to mention the potential complications from replantation. Another approach for donor site closure utilizes allogenic bone grafts [9]. This method, however, has its limitations, particularly when the inferior alveolar nerve is exposed during the bone harvesting process. The use of particulate bone substitutes in these instances might adversely affect an exposed nerve.

Our approach, which emphasizes preserving the periosteum at the retromolar donor region and ensuring layered closure, supports the concept of complete spontaneous bone healing without the need for grafting.

Historically, the reports in the relevant literature discussed the spontaneous healing of the mandibular ramus donor site. The bone deposition in this donor site was evaluated several months after the surgery. The evaluation method relied on panoramic radiographs without precise quantitative measurements. In addition, those studies also discussed the possibility of a second bone block harvesting from the same donor site [9,13,14].

Khoury and Hanser (2015) conducted a 10-year prospective study to evaluate the outcome of bone block harvesting from the mandibular ramus area. This study included 3874 bone blocks in 3328 patients with excellent long-term results. The evaluation of the donor site healing was evaluated by panoramic radiography, and the healing was apparent within 6–12 months. In addition, they reported bone regeneration after 18 months that was demonstrated through CBCT scans (in 341 patients). According to the authors, complete healing of the donor site including the external oblique line was archived in the cases in which half of the bone block had been reimplanted in the donor site [21]. 

The use of CBCT to evaluate the bone healing of the mandibular ramus donor site was reported by Diez et al. (2014). A CBCT scan was performed before the bone harvest, and at 14 days and 6 months after the surgery. The measurements were obtained in two dimensions and three dimensions. The authors reported a 76.1% bone deposition rate in the mandibular ramus donor site at 6 months after the bone block harvest. They concluded that it is possible to reharvest bone from the same site and advocated to evaluate the donor site healing after a longer period such as 12 months after the surgery [15].

Thanks to recent software advancements, our study effectively utilized Materialise’s MIS suite software for precise bone feature evaluation. This involved using CBCT scans for detailed measurements of bone density and quality, with techniques like segmentation, superimposition, and Boolean subtraction aiding in accurate volume and density assessments. These advanced tools provided us with a comprehensive understanding of the healing process, marking a significant improvement over previous methods.

Our study’s results indicate significant spontaneous healing and an improvement in bone density at the donor site following bone harvesting. The average volume of harvested bone was 606.5 mm^3^. Over time, there was a marked reduction in the residual bony defect, with 64.5% ± 4.24 healing at 6 months and 89.2% ± 2.6 healing at one year postoperatively. Additionally, the Hounsfield unit measurements showed an increase from 102.5 ± 27.8 at 6 months to 453.9 ± 91.4 at 12 months after the surgery, indicating an increase in bone density and quality over the year.

The study’s limitations include a relatively small participant number, which may affect the generalizability of our results. The limited number of CBCT scans, constrained due to patient safety concerns, hindered a comprehensive evaluation of bone healing over multiple time points. Given the ethical considerations, we refrained from conducting histologic assessments of the donor site at various time points, leaving a gap in understanding the biological aspects of donor site healing. Additionally, the one-year postoperative assessment period may not fully capture long-term outcomes, suggesting that extending the follow-up to two or three years could provide deeper insights.

In conclusion, our study demonstrates that simply preserving the periosteum and ensuring layered closure at the retromolar donor region effectively facilitates complete spontaneous bone healing, thus eliminating the need for additional grafting. Moreover, the observed volumetric and qualitative healing within a year postoperatively suggests that reharvesting from the donor site could be a viable option one year after the initial procedure.

## Figures and Tables

**Chart 1 diagnostics-14-00504-ch001:**
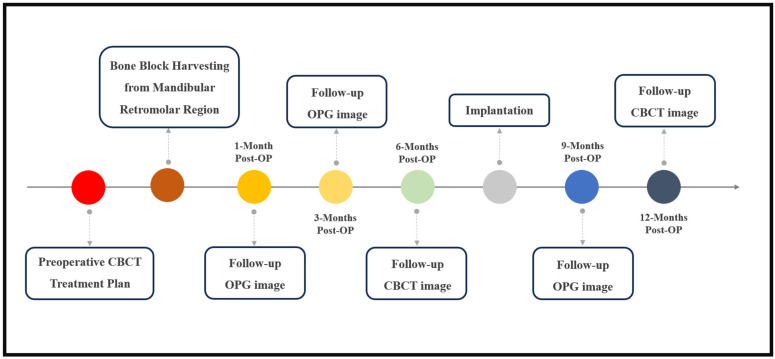
This chart depicts the sequential phases of bone augmentation and dental implant procedure, detailing the key timepoints. The process initiates with a clinical examination, preoperative CBCT, and treatment planning. The surgical intervention involves the harvest of a retromolar bone graft and ridge augmentation. The postoperative follow-up includes 1-month and 3-month assessments, comprising clinical examinations and OPG imaging. The mid-term follow-up 6 months postoperatively involves a CBCT scan and the formulation of an implantation treatment plan. The implantation procedure is then executed, followed by a 1-year post-bone augmentation CBCT scan.

**Figure 1 diagnostics-14-00504-f001:**
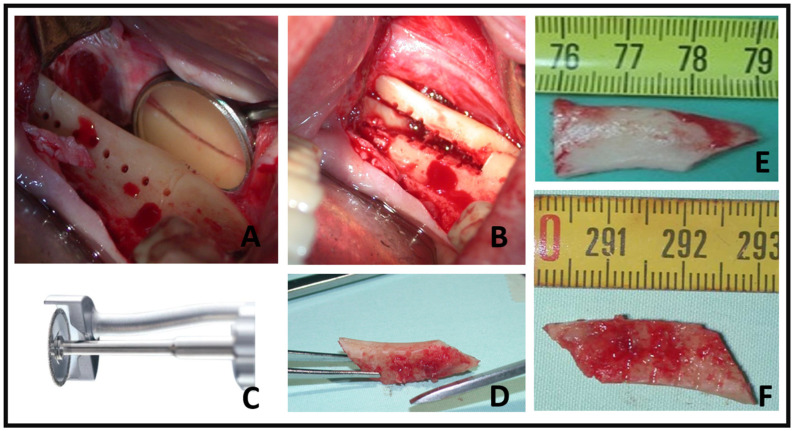
Harvesting bone block from the mandibular retromolar region (**A**) The MicroSaw handpiece was employed to execute a vertical osteotomy along the anterior and posterior borders of the external oblique ridge. The apical connection of both vertical incisions, as reflected in the mirror, was carried out, followed by crestal connections using a drill bur. (**B**) The perforations created were connected using a fine 6 mm chisel, and the block was displaced. (**C**) MicroSaw handpiece (Frios contra-angle handpiece WI-75, Dentsply). (**D**–**F**) The harvested bone blocks were presented, displaying their corticocancellous morphology from various viewing angles.

**Figure 2 diagnostics-14-00504-f002:**
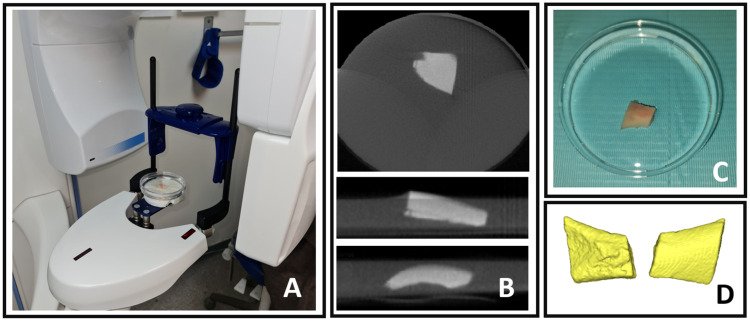
Intraoperative real-time evaluation using CBCT scan (**A**) The harvested bone, preserved in saline 0.9%, is depicted within the CBCT chamber machine. (**B**) Axial, sagittal, and coronal sections of the CBCT image. (**C**) The harvested bone block preserved in saline 0.9%. (**D**) Segmentation of the harvested graft scanning, showcasing lateral and medial views.

**Figure 3 diagnostics-14-00504-f003:**
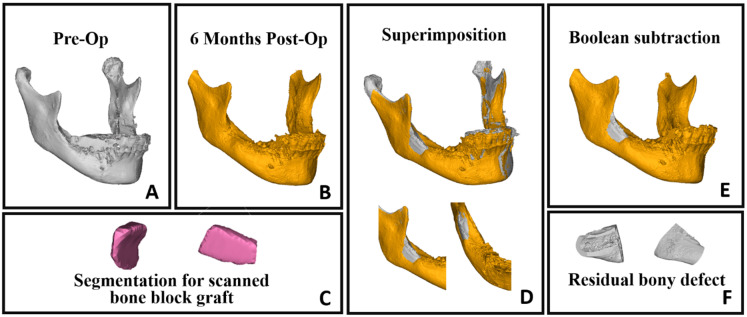
Superimposition of segmented objects and Boolean subtraction process. (**A**) Segmented mandible from preoperative CT scan. (**B**) Segmented mandible from 6-month postoperative CT scan. (**C**) Segmentation of bone block graft (medial and lateral view). (**D**) Superimposition of preoperative segmented mandible (gray) with 6-month postoperative mandible (yellow). (**E**) Boolean subtraction results showing residual bony defect (gray) on segmented 6month postoperative CT scan. (**F**) Detailed view of residual bony defect.

**Figure 4 diagnostics-14-00504-f004:**
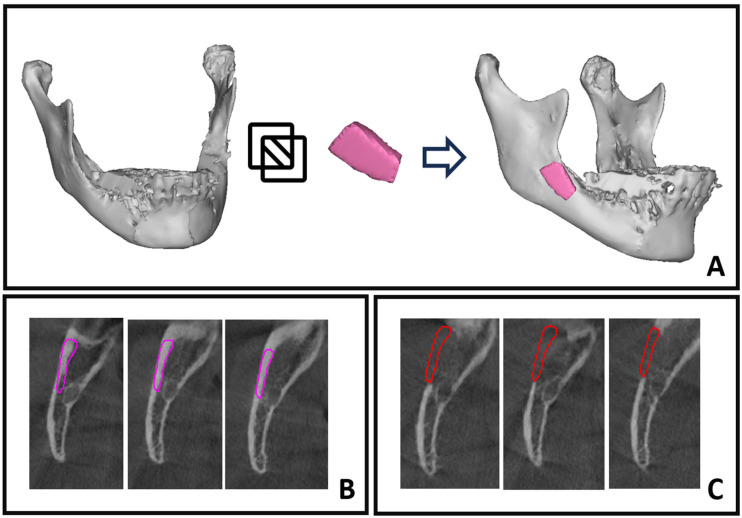
Volumetric Hounsfield evaluation for donor site. (**A**) Superimposition between a segmented mandible from preoperative CT scan (gray) and a segmented bone block from intraoperative CT scan (pink), achieved by repositioning the grafted bone block to its original place in the mandibular ramus. (**B**,**C**) display CBCT axial sections that show the boundaries of the harvested bony block object repositioned in its original place. This was utilized as a template for the volumetric HU evaluation in both the preoperative CBCT and the 6-month follow-up CBCT scan.

**Figure 5 diagnostics-14-00504-f005:**
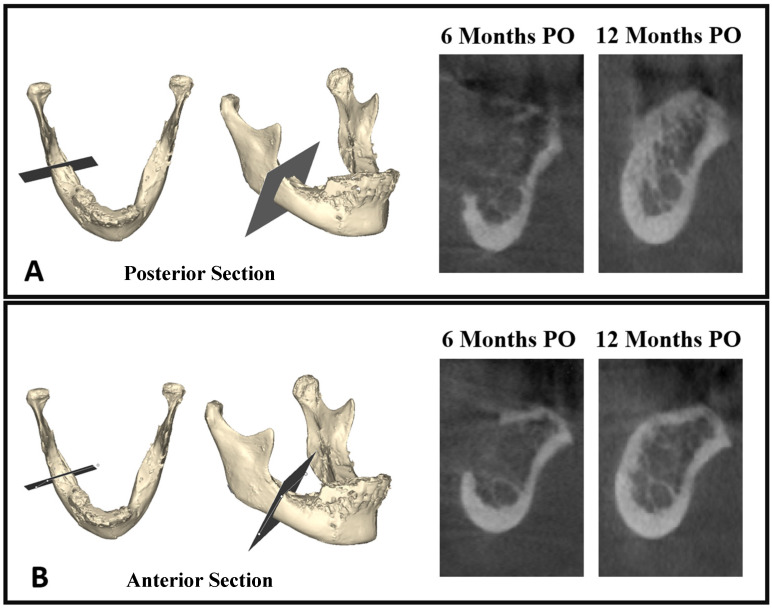
Cross-sectional view displaying bone healing at 6 months and 12 months postoperatively. Image (**A**) shows the healing at the level of the posterior section/plane, while image (**B**) depicts the healing in the anterior section/plane. These views provide a detailed comparison of bone regeneration over time in different areas of the surgical site.

**Figure 6 diagnostics-14-00504-f006:**
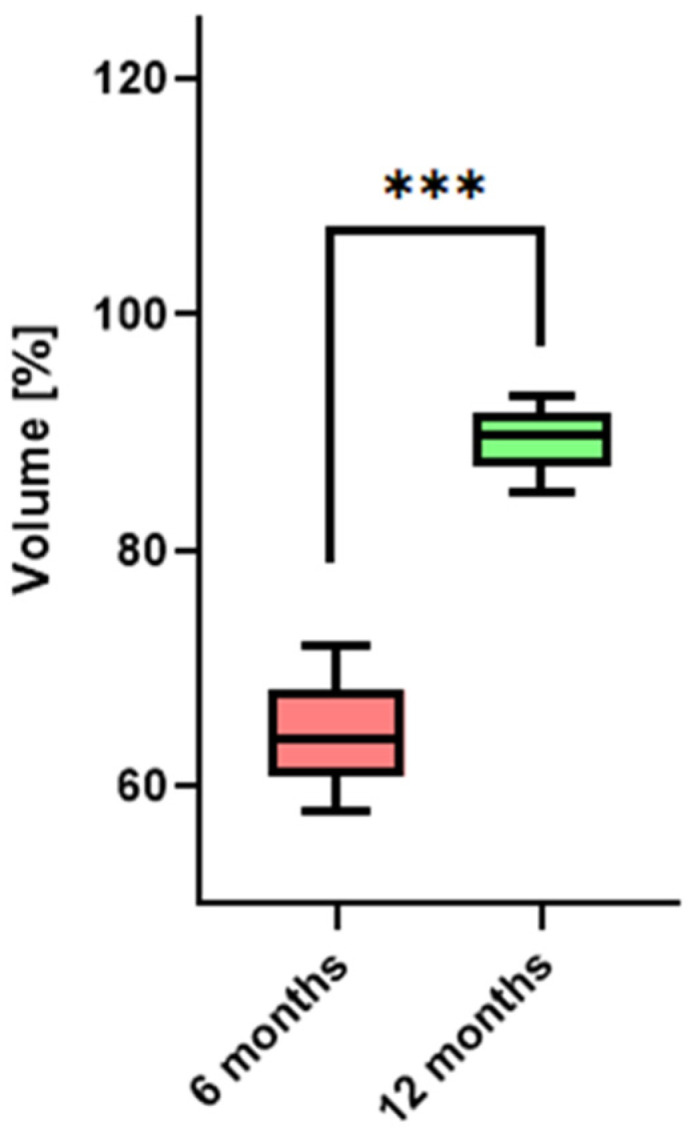
Bone healing at donor site—volumetric evaluation. The box plot illustrates the significant average volumetric healing percentage, measured at six months postoperatively (64.5% ± 4.24) and one year postoperatively (89.2% ± 2.6). *** indicates *p* values less than 0.001.

**Figure 7 diagnostics-14-00504-f007:**
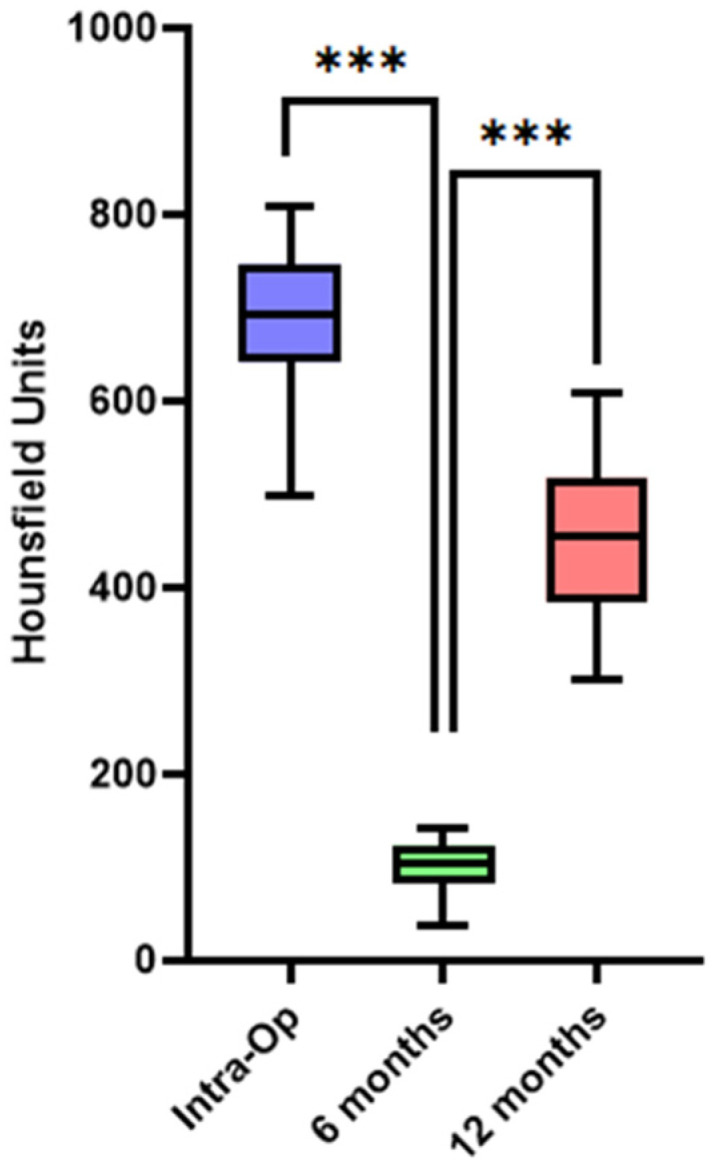
Bone healing at donor site—density/quality evaluation. The box plot shows the HU averaging at various study timepoints. The initial intraoperative HU mean was 690.3 ± 81, followed by 102.5 ± 27.8 at 6 months, and 453.9 ± 91.4 after one year. The data highlight a significant increase in average HU between 6 months and 1 year postoperatively. *** indicates *p* values less than 0.001.

**Table 1 diagnostics-14-00504-t001:** Bone healing at donor site—volumetric evaluation. The left column displays the intraoperative scanning results for the bone graft volume (mm^3^). The middle and right columns depict the residual bone defect, evaluated at six months and one-year post-surgery, respectively (mm^3^).

	Intraoperative Scanning for Bone Graft Volume	Residual Bony Defect 6 Months Postoperative	Residual Bony Defect 1 Year Postoperative
1	625.3	250.1	75.1
2	523.5	191.2	73.29
3	492.1	201.7	49.2
4	612.9	257.5	48.6
5	710.2	228.3	106.5
6	582.5	210.4	52.4
7	645.2	212.2	83.8
8	572.7	194.8	45.7
9	485.7	190.3	53.4
10	593.3	188.2	59.1
11	638.9	190.5	95.8
12	521.8	146.1	36.5
13	702.7	203.8	86.2
14	542.1	226.3	55.7
15	740.1	270.8	74.3
16	510.4	188.4	45.9
17	650.9	227.3	78.1
18	725.7	229.1	101.6
19	622.5	242.7	50.1
20	632.4	225.4	44.2

**Table 2 diagnostics-14-00504-t002:** Bone healing at donor site—density/quality evaluation. The left column presents the volumetric average of the HU evaluation for the intraoperative scans of the bone block grafts. The middle and right columns depict the average HU evaluation based on 6-month and one-year follow-up CBCT scans, respectively.

	Intraoperative Scanning for Bone Graft	CBCT 6 Months Follow-Up Scan	CBCT 12 Months Follow-Up Scan
1	752	124	561
2	712	56	457
3	643	113	503
4	784	88	610
5	578	110	364
6	801	78	580
7	595	38	312
8	735	140	455
9	683	124	423
10	810	133	593
11	498	98	301
12	654	128	417
13	727	141	472
14	674	89	393
15	729	122	475
16	641	76	443
17	705	99	524
18	795	117	485
19	658	81	328
20	633	96	382

## Data Availability

The data that support the findings of this study are available from the corresponding author upon reasonable request.

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
