# Peer review of "Computer-Assisted Evaluation Confirms Spontaneous Healing of Donor Site One Year following Bone Block Harvesting from Mandibular Retromolar Region—A Cohort Study"

_diagnostics, 2024, doi:10.3390/diagnostics14050504_

Round 1

Reviewer 1 Report

Comments and Suggestions for Authors

The following points should be addressed before it can be further considered for publication.

Title

1- The observation period (i.e., one-year follow-up) should appear in the title.

2- The type of healing should be clearly stated. In other words, "Complete healing" in terms of what? 

3- Please use "CBCT-based analysis" instead of "Computer-assisted evaluation". You are using 3D techniques and not computer software.

4- The phrase "A cohort study" should be added at the end of the title.

Abstract

5- The "Introduction" section should be shortened, and the "Materials and Methods" section should be expanded. The Results section should contain some numbers and P-values. There should be some results in the "Results" section.

Introduction

6- Line 41: When there are two citations, the numbers should be combined within two parentheses. 

7- Line 54. When there are three citations, their numbers should be combined within two parentheses.

Materials and Methods

8- What is the study design for this work?

9- Did you perform a priori sample size calculation?

9- Line 84: Please make the first letter or the word "Six" a small letter.

8- Is it ethical to have 3 CBCTs within one year? Is it safe?

9- In Chart 1, please correct the abbreviation OPJ to become "OPG". I think that the OPJ has been used incorrectly. 

Results

10- The authors present their raw data in tables. However, we would like to see some descriptive statistics. Also, the significance of changes between the beginning and six months should be given. The significance of changes between six and 12 months should also be given.

Comments on the Quality of English Language

The English language is good. Some grammatical errors should be corrected.

Some mistakes appeared regarding the capitalization of some words.

Author Response

Dear Review Editor,

We sincerely thank you for taking the time to review our manuscript and for providing constructive feedback. Your comments and suggestions are invaluable in enhancing the quality and clarity of our work.

We have carefully considered each point raised in your review and have addressed them individually in our revised manuscript to ensure that all your concerns and requests are thoroughly attended to. Below, we provide detailed responses to each of your comments, explaining the changes we have made to the manuscript accordingly.

- Title

1- The observation period (i.e., one-year follow-up) should appear in the title.

The title was changed to-

Computer-Assisted Evaluation Confirms Spontaneous Healing of Donor Site One Year Following Bone Block Harvesting from Mandibular Retromolar Region - A Cohort Study

2- The type of healing should be clearly stated. In other words, "Complete healing" in terms of what?

"The term 'Complete Healing' was removed from the revised title."

3- Please use "CBCT-based analysis" instead of "Computer-assisted evaluation". You are using 3D techniques and not computer software.

We value your suggestion on terminology. However, we advocate for retaining "computer-assisted" to describe our methodology, given our extensive use of MIS software for segmentation, alignment, Boolean operations, and HU evaluations, which extends beyond mere CBCT image analysis. This term accurately reflects our comprehensive approach, aligning with the accepted literature on software-based analysis from DICOM images.

4- The phrase "A cohort study" should be added at the end of the title.

“A cohort study” was included in the revised title.

- Abstract

5- The "Introduction" section should be shortened, and the "Materials and Methods" section should be expanded. The Results section should contain some numbers and P-values. There should be some results in the "Results" section.

Following your guidance, we've concisely revised the abstract: the introduction is now more compact, providing essential context succinctly. We've extended the materials and methods section for a detailed methodology outline and enhanced the results section with more comprehensive data and analysis.

- Introduction

6- Line 41: When there are two citations, the numbers should be combined within two parentheses. 

7- Line 54. When there are three citations, their numbers should be combined within two parentheses.

Requested changes were made.

- Materials and Methods

8- What is the study design for this work?

Thank you very much for your question. In the current work, patients who underwent an alveolar ridge augmentation with a retromolar bone graft were followed for up to one year post-operatively. CBCT scans were used to evaluate the bone healing at the harvest site. Therefore, an observational prospective cohort study was performed.

9- Did you perform a priori sample size calculation?

Thank you for your question and the opportunity to clarify.

The sample size calculation was based on paired sample T-test, to achieve a power of 90 % and a significance value of 0.5. The minimal difference in HU values between time points considered as a significant result was estimated to be 100 HU with a standard deviation of 140. Thus, the calculated sample size was 23 (SPSS Statistics software version 27.0, IBM). We based our sample design on previous studies in which similar cohorts were followed for jaw healing, in which significantly larger mean differences in HU values emerged between time points [1][2].

Das, S., Jhingran, R., Bains, V.K., Madan, R., Srivastava, R. and Rizvi, I., 2016. Socket preservation by beta-tri-calcium phosphate with collagen compared to platelet-rich fibrin: A clinico-radiographic study. European journal of dentistry, 10(02), pp.264-276.

Cavdar FH, Keceli HG, Hatipoglu H, Demiralp B, Caglayan F. Evaluation of Extraction Site Dimensions and Density Using Computed Tomography Treated With Different Graft Materials: A Preliminary Study. Implant Dentistry Journal. 2017 Apr;26(2):270-274.

9- Line 84: Please make the first letter or the word "Six" a small letter.

Requested changes were made.

8- Is it ethical to have 3 CBCTs within one year? Is it safe?

In our study, we significantly reduced radiation exposure by adopting an innovative approach where the intraoperative CBCT focused on the bone graft instead of the patient. This strategy eliminated the need for multiple CBCT scans within a single year. The preoperative CBCT mostly was done up to a year prior to surgery, ensuring that participants did not undergo three CBCT scans within a year. Furthermore, the CBCT scans conducted in-house for 6 months and one year postoperatively were performed with strict safety measures and scanning protocols designed to minimize radiation exposure.

9- In Chart 1, please correct the abbreviation OPJ to become "OPG". I think that the OPJ has been used incorrectly. 

Requested changes were made.

- Results

10- The authors present their raw data in tables. However, we would like to see some descriptive statistics. Also, the significance of changes between the beginning and six months should be given. The significance of changes between six and 12 months should also be given.

In response to the request for a broader statistical analysis, we have expanded our analysis in the revised manuscript. This updated analysis now includes and emphasizes the statistical significance of the observed changes from the baseline to six months, as well as from six months to one year postoperatively. These enhancements provide a clearer understanding of the temporal dynamics and the clinical impact over the study period.

- Comments on the Quality of English Language: Some grammatical errors should be corrected. Some mistakes appeared regarding the capitalization of some words.

Requested changes were made.

We believe that these revisions have significantly improved our manuscript, making it a stronger contribution to the field. We are grateful for the opportunity to refine our work with your guidance and look forward to any further suggestions you may have.

Thank you once again for your thoughtful review and valuable insights.

Warm regards,

Shadi Daoud

Faris Kablan

Reviewer 2 Report

Comments and Suggestions for Authors

Dear Authors the manuscript is well-structured and clear.

Citations could be implemented relative to the topic.

The aim of this study is to be appropriately declared, but the authors should better specify its rational basis.

Introduction: line 66, please add from recent literature such as doi: 10.23736/S0026-4970.17.04003-1

Materials and Methods: This method was appropriate. The authors should specify the sample size calculation.

Result section: should be reorganized by moving some sentences of Figures in the text (too much text in the figure legends).

Figures 6A and 7A are redundant in the contents of Tables 1 and 2. Please reduce and summarize this manuscript.

When you explain the necessity of bone augmentation in patients with compromised alveolar bone, you should briefly cite other less invasive grafts that are often used.

Discussion: Please add a comparison (advantages and disadvantages) between autogenous bone grafts and xenografts, and in which cases you should suggest them.

line 294 I would also add possible alternative techniques such as the possibility of inserting short implants or free fibula flaps.

Please add the limitation of the study at the end of the discussion part

Please add content regarding the biological mechanisms of spontaneous bone healing at the donor site.

Please discuss the last point regarding the necessity of re-harvesting one year after the initial procedure.

The conclusions are consistent with the evidence and arguments presented.

Please evaluate the ethics and data availability statements to ensure that they are adequate.

Author Response

Dear Review Editor,

We sincerely thank you for taking the time to review our manuscript and for providing constructive feedback. Your comments and suggestions are invaluable in enhancing the quality and clarity of our work.

We have carefully considered each point raised in your review and have addressed them individually in our revised manuscript to ensure that all your concerns and requests are thoroughly attended to. Below, we provide detailed responses to each of your comments, explaining the changes we have made to the manuscript accordingly.

-  The authors should better specify its rational basis.

The rationale behind this study stems from the observation that the mandibular ramus serves as a main intraoral harvest site for bone grafting, showcasing a high success rate for enduring alveolar ridge augmentation, extending to complete jaw augmentation or comprehensive bone reconstruction. The preservation of the donor site, alongside the potential for reharvesting from the same location, presents significant benefits. It offers a less invasive alternative to more morbid surgeries that require extraoral bone harvesting, thereby reducing patient morbidity and enhancing the recovery process.

- Introduction: line 66, please add from recent literature such as doi: 10.23736/S0026-4970.17.04003-1

We have added the recommended reference into the revised manuscript as suggested.

- The authors should specify the sample size calculation.

Thank you for your question and the opportunity to clarify.

The sample size calculation was based on paired sample T-test, to achieve a power of 90 % and a significance value of 0.5. The minimal difference in HU values between time points considered as a significant result was estimated to be 100 HU with a standard deviation of 140. Thus, the calculated sample size was 23 (SPSS Statistics software version 27.0, IBM). We based our sample design on previous studies in which similar cohorts were followed for jaw healing, in which significantly larger mean differences in HU values emerged between time points [1][2].

Das, S., Jhingran, R., Bains, V.K., Madan, R., Srivastava, R. and Rizvi, I., 2016. Socket preservation by beta-tri-calcium phosphate with collagen compared to platelet-rich fibrin: A clinico-radiographic study. European journal of dentistry, 10(02), pp.264-276.

Cavdar FH, Keceli HG, Hatipoglu H, Demiralp B, Caglayan F. Evaluation of Extraction Site Dimensions and Density Using Computed Tomography Treated With Different Graft Materials: A Preliminary Study. Implant Dentistry Journal. 2017 Apr;26(2):270-274.

- Result section: should be reorganized by moving some sentences of Figures in the text (too much text in the figure legends).

Suitable changes have been made to the manuscript in accordance with your suggestions.

- Figures 6A and 7A are redundant in the contents of Tables 1 and 2. Please reduce and summarize this manuscript.

We have removed Figures 6A and 7A and summarized the remaining figure legends for clarity and conciseness.

- When you explain the necessity of bone augmentation in patients with compromised alveolar bone, you should briefly cite other less invasive grafts that are often used.

A brief summary of bone augmentation techniques has been incorporated into the introduction section, offering an overview of the methods utilized in this field.

- Discussion: Please add a comparison (advantages and disadvantages) between autogenous bone grafts and xenografts, and in which cases you should suggest them.

We appreciate the interest in xenograft augmentation; however, our study specifically investigates bone healing at the donor site without graft material, placing such discussions beyond our current scope.

- line 294 I would also add possible alternative techniques such as the possibility of inserting short implants or free fibula flaps.

Thank you for your valuable suggestion regarding the inclusion of alternative techniques such as short implants and free fibula flaps. As we have discussed in the article, autogenous bone grafts are considered the gold standard for grafting due to their proven long-term results. While short implants are indeed an alternative, literature suggests their long-term survival and success rates are still under scrutiny and not as well-established. Regarding the free fibula flap, it is indeed a viable option but is generally reserved for cases of severe bone defects due to its higher morbidity. We appreciate your insights and have aimed to focus on the most widely accepted and successful practices within the scope of our study.

- Please add the limitation of the study at the end of the discussion part

Study limitations were added at the end of the discussion part.

- Please add content regarding the biological mechanisms of spontaneous bone healing at the donor site.

Thank you for emphasizing the need to elucidate the biological healing process at the donor site. As detailed in our introduction and methodology, we took great care to preserve and suture the periosteum layer, understanding its pivotal role in enhancing healing. The healing mechanism fundamentally relies on osteogenesis, driven by both the bone medulla and the carefully preserved periosteum. Recognized as a vital source of osteoprogenitor cells, the periosteum is instrumental in mending bone defects. Additionally, the supportive osteogenesis effect of the exposed medulla post-harvesting plays a significant role. To comprehensively address this point, we have included a summarized note in the article. This addition explicitly outlines the contribution of these key sources to the healing process of the donor site.

- Please discuss the last point regarding the necessity of re-harvesting one year after the initial procedure.

As highlighted in both the introduction and discussion sections, our study emphasizes the mandibular ramus as a primary intraoral harvest site for bone grafting, noted for its high success rate in alveolar ridge augmentation. The capability to preserve the donor site, coupled with the potential for reharvesting from the same location, offers considerable advantages. Our findings demonstrate that one year post-operation, the donor site retains sufficient bone volume and quality, suggesting its viability for reharvesting. This underscores the effectiveness of the mandibular ramus not only as a reliable source for initial grafting but also for subsequent procedures, reinforcing the site's value in dental reconstructive strategies.

- Please evaluate the ethics and data availability statements to ensure that they are adequate.

The study received ethical approval from the Israeli Helsinki Committee system at the Galilee Medical Center in Nahariya, Israel. Approval number 0149-21-NHR

We believe that these revisions have significantly improved our manuscript, making it a stronger contribution to the field. We are grateful for the opportunity to refine our work with your guidance and look forward to any further suggestions you may have.

Thank you once again for your thoughtful review and valuable insights.

Warm regards,

Shadi Daoud

Faris Kablan

Round 2

Reviewer 1 Report

Comments and Suggestions for Authors

Thanks for addressing all of my raised points in my preview review of the paper.